# Production and Optimization of Biosurfactant Properties Using *Candida mogii* and Licuri Oil (*Syagrus coronata*)

**DOI:** 10.3390/foods13244029

**Published:** 2024-12-13

**Authors:** Peterson F. F. da Silva, Renata R. da Silva, Leonie A. Sarubbo, Jenyffer M. C. Guerra

**Affiliations:** 1Centro de Tecnologia e Geociências, Programa de Pós-Graduação em Engenharia Química, Departamento de Engenharia Química, Universidade Federal de Pernambuco (UFPE), Avenida Professor Moraes Rego, n. 1235, Cidade Universitária, Recife 50670-901, PE, Brazil; peterson.fsilva@ufpe.br; 2Rede Nordeste de Biotecnologia (RENORBIO), Universidade Federal Rural de Pernambuco (UFRPE), Rua Dom Manuel de Medeiros, s/n—Dois Irmãos, Recife 52171-900, PE, Brazil; renatabiology2015@gmail.com; 3Instituto Avançado de Tecnologia e Inovação (IATI), Rua Potyra, n. 31, Prado, Recife 50070-280, PE, Brazil; leonie.sarubbo@unicap.br; 4School of Technology and Communication, Catholic University of Pernambuco (UNICAP), Rua do Príncipe, n. 526, Boa Vista, Recife 50050-900, PE, Brazil

**Keywords:** microbial surfactant, central composite design, optimization, antibacterial, yeast

## Abstract

Optimizing biosurfactant (BS) production is key for sustainable industrial applications. This study investigated BS synthesis by *Candida mogii* using licuri oil, a renewable carbon source rich in medium-chain fatty acids. Process optimization was conducted via central composite design (CCD), adjusting concentrations of licuri oil, glucose, NH_4_NO_3_, and yeast extract. The predictive model achieved an R^2^ of 0.9451 and adjusted R^2^ of 0.8812. Under optimized conditions, *C. mogii* lowered water surface tension from 71.04 mN·m^−1^ to 28.66 mN·m^−1^, with a critical micelle concentration (CMC) of 0.8 g·L^−1^. The biosurfactant displayed high emulsification indices, exceeding 70% for canola, licuri, and motor oils, suggesting strong potential as an industrial emulsifier. FTIR and NMR analyses confirmed its glycolipid structure. Bioassays showed no toxicity to *Lactuca sativa* seeds, ensuring environmental safety, while antimicrobial tests demonstrated efficacy against *Staphylococcus aureus* and *Escherichia coli*, indicating its suitability as a biocidal agent. This work positions *C. mogii* BS from licuri oil as a promising alternative for bioremediation, biotechnology, and antimicrobial uses.

## 1. Introduction

Surfactants play a central role in industry due to their amphipathic properties, characterized by the presence of hydrophilic and hydrophobic segments within their molecules [1]. This amphiphilic structure is a key factor in reducing surface and interfacial tension between immiscible substances, making them highly applicable across various sectors [2,3]. Based on their origin, they are classified as natural or synthetic. Synthetic surfactants, derived from fossil fuels, raise concerns due to their potential adverse effects on health and the environment, associated with toxicity and low biodegradability [4]. In this context, biosurfactants (BS) emerge as a sustainable alternative to address these issues [5].

Defined as natural products obtained from microbial metabolism [6], biosurfactants have seen remarkable growth, with the market projected to reach USD 6.3 billion by 2026 [7]. These microbiologically derived compounds offer advantages such as biodegradability, low toxicity, selectivity, and production from renewable and low-cost raw materials [8].

Sourced from various organisms, including bacteria, fungi, and yeasts, biosurfactants exhibit a wide range of chemical structures and distinct properties [9]. In the current scenario, yeasts have been used for surfactant production due to their GRAS (Generally Recognized as Safe) status, enabling broad applications [10]. Within this group, the *Candida* genus has been extensively explored for its industrial significance, with BS produced by *C. tropicalis* used in oil removal from contaminated areas [11], and those from *C. bombicola* and *C. utilis* used as food additives [12,13].

However, *Candida mogii*, a yeast with well-documented metabolic versatility, remains unexplored for biosurfactant production. Known for its efficient xylitol production from xylose and its adaptability to diverse fermentation conditions [14,15], it presents a compelling candidate for biosurfactant synthesis.

Currently, high production costs result in low economic competitiveness, due to the use of expensive substrates and inefficient product recovery methods. Optimizing the production medium composition is a promising approach, potentially reducing costs by up to 40% [16,17,18], leading to the maximization of desirable factors such as yield and surface tension reduction [19]. For instance, the application of Central Composite Design (CCD) reduced surface tension from 61 to 34 mN·m^−1^ for BS produced by *Bacillus mycoides*, while the yield of the surfactant produced by *C. lipolytica* increased from 0.59 g·L^−1^ to 7.27 g·L^−1^ using a 2^4^ factorial design [20,21].

New natural carbon sources are often investigated to understand how microorganisms respond to them and thus improve yields [22]. Media containing vegetable oils present promising substrates [23]. Within this context lies licuri oil (*Syagrus coronata*), an edible oil traditionally consumed in Northeastern Brazil, which is primarily composed of medium-chain fatty acids, making it an excellent candidate for producing lipid-derived biomolecules. Beyond its nutritional value, licuri oil is a key product of local agroecosystems and rural livelihoods, with potential to generate added value through biotechnological applications [18]. Souza et al. [24] demonstrated that the oil in question was non-toxic, making it safe for pharmacological and food applications.

Considering the need to research new substrates and producing strains for the viability of biomolecule production, this study aimed to explore *Candida mogii* as a novel biosurfactant-producing strain by optimizing its production medium using licuri oil, a renewable and edible carbon source, while evaluating its phytotoxicity and antimicrobial activity.

## 2. Materials and Methods

### 2.1. Materials

The chemical reagents and culture media used in this study were of analytical grade. The crude oil, obtained via mechanical pressing of *Syagrus coronata* almonds, was provided by the Cooperativa de Produção da Região do Piemonte da Diamantina (COOPES), located in the city of Capim Grosso, Bahia, Brazil (11°22′51″ S, 40°00′46″ W).

### 2.2. Maintenance and Growth of the Microorganism

The yeast *Candida mogii* UFPEDA 3968 was obtained from the culture collection of the Mycology Department at the Federal University of Pernambuco. The microorganism was maintained at a temperature of 5 ± 1 °C on a yeast mold agar (YMA) medium composed of (% *w*/*v*) yeast extract, 0.3%; malt extract, 0.3%; tryptone, 0.3%; glucose, 1.0%; and agar, 2.0% [25].

### 2.3. Yeast Inoculation

The yeast inoculum standardization was performed by transferring the culture to a tube containing the YMA medium to obtain a young culture at 28 ± 1 °C. Subsequently, the sample was placed in a 250 mL Erlenmeyer flask with 100 mL of yeast mold broth (YMB) medium composed of (% *w*/*v*) yeast extract, 0.3%; malt extract, 0.3%; tryptone, 0.3%; and glucose, 1.0%. It was then incubated aerobically with orbital shaking at 200 rpm at 28 ± 1 °C for 24 h in a C25KC shaker incubator (New Brunswick Scientific, Edison, NJ, USA).

### 2.4. Biosurfactant Production Medium

The culture medium for biosurfactant production used licuri oil and glucose in a mineral base containing (% *w*/*v*) NH_4_NO_3_, 0.1%; KH_2_PO_4_, 0.01%; MgSO_4_∙7H_2_O, 0.5%; FeCl_3_, 0.01%; and NaCl, 0.01%, with the pH adjusted to 5.7, in 250 mL Erlenmeyer flasks containing 100 mL of medium [26]. The incubation conditions were maintained for 168 h at 200 rpm, using a 1% inoculum with a concentration of 10^7^ cells/mL, quantified in a Neubauer chamber.

To identify the ideal production conditions, a Central Composite Rotational Design (CCRD) was used, configured in the software Design Expert Stat-Ease^®^ 360 (State-Ease Inc., Minneapolis, MN, USA), considering the following factors: licuri oil concentration (A), glucose (B), ammonium nitrate (C), and yeast extract (D). The factor levels varied between minimum (−1), central (0), maximum (+1), and axial points (±α), with the specific concentrations indicated in Table 1. Biosurfactant production was evaluated by the ability to reduce surface tension (ST_red_) in cell-free culture, considered the dependent variable. The total number of experiments considered in the approach was 27, including 3 repetitions for the central point, to account for the magnitude of possible random errors.

The dependent variable STred was calculated from the following relation:ST_red_ = ST_w_ − ST_exp_,(1)
where ST_w_ is the measurement of the surface tension of water (standard procedure during equipment calibration), and ST_exp_ is the measured surface tension obtained by the equipment.

The results were analyzed to develop an appropriate model for surface tension reduction. An analysis of variance (ANOVA) was performed to assess the significant impact of the independent factors on biosurfactant production, using F and *p* values. The goodness of fit of the polynomial model was determined by the R^2^ and adjusted R^2^ coefficients. The software was used to identify factor-level combinations that optimized surface tension reduction, employing the quadratic model:(2)$$Y=\beta_{0}+\sum_{i=1}^{4}\beta_{i}X_{i}+\sum_{i=1}^{4}\beta_{ii}X_{i}^{2}+\sum_{i=1}^{3}\sum_{j=1+1}^{4}\beta_{ij}X_{i}X_{j}+\epsilon ,$$
where Y is the predicted response (ST_red_), X represents the independent factors, i.e., (X_1_ = A, X_2_ = B, X_3_ = C, X_4_ = D), β_0_ is the intercept, β_i_ are the coefficients for individual factors, β_ii_ are the quadratic coefficients, and β_ij_ are the coefficients for the interactions between factors (for example, β_12_ for AB).

### 2.5. Surface Tension Measurement

The surface tension measurements were performed in cell-free broth, after removing the cells by centrifugation at 4500 rpm for 10 min in a centrifuge (Rotina 420R, Hettich Zentrifugen, Tuttlingen, Germany) at 4 ± 1 °C. For this procedure, the Du–Nuoy ring method was used with a Sigma 70 tensiometer (KSV Instruments LTD, Helsinki, Finland) at room temperature (25 ± 1 °C).

### 2.6. Growth Curve

To determine the optimal production time, the growth profile was analyzed based on the optimized composition that resulted in the greatest reduction in surface tension (ST_red_). The culture was performed in 1 L Erlenmeyer flasks containing 500 mL of the production medium, under orbital shaking at 200 rpm, for a total of 216 h at 28 ± 1 °C. Samples were collected at 0, 2, 6, 12, 24, and up to 216 h (at regular 24 h intervals after the first day of fermentation) to determine biomass (g·L^−1^), surface tension (mN·m⁻^1^), and yield (g·L^−1^) [27]. The determinations were performed in triplicate.

### 2.7. Emulsification Index (E_24_)

The emulsification index for the composition that showed the best reduction in surface tension was determined against canola, licuri, soybean, diesel, kerosene, and used motor oils, following the method described by Cooper and Goldenberg [28]. For this, 2 mL of hydrocarbon was added to 2 mL of the cell-free metabolic liquid in a test tube, followed by shaking for 2 min. The stability of the emulsion was evaluated after 24 h and calculated as follows:(3)$$E_{24}=\frac{h_{e}}{H}\times 100,$$
where *h_e_* denotes the height of the emulsion column, and *H* represents the total height of the column formed by the mixture of the two liquids; the result is expressed as a percentage. A higher E_24_ percentage indicates a more stable emulsion, while a lower one suggests reduced stability.

### 2.8. Biosurfactant Isolation

The biosurfactant was isolated from the non-centrifuged culture medium, i.e., the entire broth obtained after 168 h of fermentation, using ethyl acetate in a 1:4 (*v*/*v*) ratio with the culture medium. The organic phase was collected, and the process was repeated two more times with the aqueous phase to ensure maximum extraction. Then, the organic phase was centrifuged for 15 min at 4500 rpm and filtered under vacuum. The filtrate was returned to the separation funnel, with the addition of saturated sodium chloride (NaCl) to facilitate the removal of the residual aqueous phase. Anhydrous magnesium sulfate (MgSO_4_) was added to the organic phase, followed by filtration and drying in an oven at 50 ± 1 °C [17].

### 2.9. Critical Micelle Concentration

The critical micelle concentration (CMC) was determined by measuring the surface tension of dilutions of the isolated biosurfactant in distilled water until a constant surface tension was reached. The CMC value was obtained from the graph relating surface tension and biosurfactant concentration, expressed in g·L^−1^ [12].

### 2.10. Determination of Ionic Charge and Particle Size

The determination of the ionic charge and particle size of the biosurfactant was conducted using the Malvern Zetasizer Nano ZS90 (Malvern Instruments Inc., Worcestershire, UK), which allows for accurate measurements of the zeta potential and the average particle diameter in the sample.

### 2.11. Fourier Transform Infrared Spectroscopy (FT-IR)

FT-IR analysis was performed using the IRSpirit spectrophotometer (Shimadzu, Kyoto, Japan) with a total attenuated reflectance accessory (QATR-S). The measurements were made in percentage transmittance (%T) mode with Happ-Genzel apodization, covering the mid-infrared range of 400 to 4000 cm^−1^. Each spectrum was obtained with 42 scans at a resolution of 16 cm^−1^.

### 2.12. Nuclear Magnetic Resonance Spectroscopy (NMR)

The analyses of ^1^H and ^13^C NMR were performed on a Bruker Avance-NEO-600 NMR spectrometer (Bruker Corporation, Billerica, MA, USA), operating at 500 MHz for the hydrogen nucleus. Twenty milligrams of the isolated biosurfactant were prepared, dissolved in 500 µL of deuterated chloroform (CDCl_3_; Sigma-Aldrich, Taufkirchen, Germany), maintaining the temperature at 298.1 K. The chemical shift scale (δ) was referenced in ppm, using tetramethylsilane (TMS) as an internal standard.

### 2.13. Phytotoxicity Assay

The phytotoxicity of the biosurfactant was evaluated in a static assay, focusing on the seed germination and root growth of *Lactuca sativa* (lettuce), following the methodology described by Tiquia and collaborators [29]. Solutions of both crude and isolated biosurfactant were prepared in distilled water at concentrations of ^1^/_2_ CMC (0.25%), 1 CMC (0.5%), and 2 CMC (1.0%). The assay was conducted in sterilized Petri dishes lined with filter paper, where seeds, previously treated with sodium hypochlorite, were placed (10 seeds per dish) along with 5 mL of each test solution. After five days of incubation in the dark, the germination rate and root lengths were measured, as described in the study:(4)$$Relative\;Seed\;Germination\;(RSG,\%)\;=\;\left(\frac{G_{e}}{Gc}\right)\times 100$$
(5)$$Relative\;Root\;Length\;(RRL,\%)\;=\;\left(\frac{L_{e}}{Lc}\right)\times 100,$$
(6)$$Germination\;Index\;=\;\left(\frac{RSG\times RRL}{100}\right),$$
where Ge, Gc, Le, and Lc represent the quantities of germinated seeds and the root lengths in the tested samples (subscript e) and control (subscript c), respectively.

### 2.14. Antimicrobial Activity Potential

A disk diffusion assay was conducted to evaluate the antibacterial activity of the biosurfactant. Gram-negative (*Escherichia coli*) and Gram-positive (*Staphylococcus aureus*) bacteria were tested at a concentration of 10^8^ CFU·mL^−1^. Filter paper disks impregnated with the biosurfactant in solutions of different concentrations: 0.4 g·L^−1^ (½ CMC); 0.8 g·L^−1^ (1 CMC); 1.6 g·L^−1^ (2 CMC); fermentation metabolic liquid and sterile water (negative control) were applied to Mueller–Hinton Agar (Merck, Darmstadt, Germany) plates. After 24 h of incubation at 30 ± 1 °C, the inhibition zones were measured in duplicate, following the CLSI protocol [30,31].

### 2.15. Statistical Analysis

The data were statistically analyzed using Statistica^®^ software (version 12), followed by an analysis of variance (ANOVA). All experiments were performed in triplicate, and the results were expressed as means ± standard deviations (SD), with a 95% confidence interval.

## 3. Results and Discussion

### 3.1. Biosurfactant Production

Table 2 presents the experimental setup established by the Design Expert Stat-Ease^®^ 360 software, including the surface tension measurements and the experimental and predicted values of the response variable (ST_red_). The obtained data highlight the potential of *Candida mogii* as a biosurfactant producer, demonstrating a substantial reduction in the surface tension of water, from 71.179 ± 0.332 mN·m^−1^ to values ranging between 28.397 ± 0.101 and 48.848 ± 0.216 mN·m^−1^. This variation range underscores the significant influence of the substrate composition on the biosurfactant production by the studied strain.

The effectiveness of a biosurfactant is determined by its ability to reduce surface tension; a greater reduction indicates higher effectiveness [32]. The experimental design indicated configuration 12, composed of (%) licuri oil (5), glucose (4), NH_4_NO_3_ (0.5), and yeast extract (0.5), as the composition that most effectively reduced the surface tension of the medium, reaching a minimum value of 28.397 ± 0.508 mN·m^−1^ (ST_red_ of 42.782 mN·m^−1^). This result highlighted the yeast as comparable to other *Candida* strains known for their efficient reduction in surface tension, such as *C. tropicalis*, which was able to reduce the surface tension of water from 72 mN·m^−1^ to 30 mN·m^−1^ [11]; *C. bombicola*, which achieved 29 mN·m⁻^1^ [33]; and *C. lipolytica*, which showed an excellent value of 25 mN·m^−1^ [27].

The efficiency in surface tension reduction (ST_red_) as a function of the independent variables was modeled with a second-order polynomial equation, based on 15 terms. The equation, adjusted to the experimental data, considered the coded variables A, B, C, and D, representing different concentrations of medium components. This approach allowed for the identification of the individual and interactive influence of each variable on the response, providing a predictive model to optimize the formulation, as shown:ST_red_ = 31.291 + 0.134A − 0.519B + 0.154C + 3.409D − 1.149AB + 0.595AC + 1.810AD − 1.433BC − 1.981BD + 0.766CD − 0.316A^2^ + 0.065B^2^ + 0.102C^2^ − 0.597D^2^, (7)

The analysis of variance (ANOVA) for the quadratic model of surface tension reduction (ST_red_) is presented in Table 3 and revealed that the model was highly significant, with an F-value of 14.78 and *p* < 0.0001, indicating a strong fit to the experimental data. Among the terms, the main factor D (yeast extract) and the interactions AB, AC, AD, BC, and BD showed significant influence (*p* < 0.05), suggesting that these terms had a considerable impact on the response. The lack of fit absence (*p* = 0.261) confirmed the model’s adequacy, ensuring accurate data representation and reliable predictions.

The fit statistics reinforced the adequacy of the model, with R^2^ = 0.9451 and R^2^_adj_ = 0.8812, indicating a robust explanation of the data variability. The precision level of 17.924, well above the minimum desired (4), suggested an excellent signal-to-noise ratio, indicating that the model was reliable for exploring the design space [34]. As seen in Figure 1, the normal probability plot of externally studentized residuals shows points aligning closely with the red line, indicating residuals reasonably followed a normal distribution, with minimal outliers [35].

Figure 2 illustrates the Pareto chart with the main effects of each factor influencing the reduction in surface tension. Negative values for the interactions BD, BC, and AB suggested that the combination of these factors decreased their effect on the response variable. In other words, when variables B (glucose) and D (yeast extract), B and C (ammonium nitrate), or A (licuri oil) and B interacted, the combined effect of these interactions resulted in a smaller reduction in surface tension than if these variables acted individually or in other combinations. On the other hand, the positive values observed for variables D and AD indicated that these factors had an effect that enhanced the effectiveness of the surface tension reduction when interacting. This means that yeast extract (D) alone and in interaction with licuri oil (A) contributed positively to the desired response, potentially maximizing the surface tension reduction and standing out as favorable conditions for the experiment’s goal. It can also be noted that the quadratic terms (Q) did not show significant effects on surface tension reduction, a conclusion that could be directly verified from the ANOVA table.

The goal of optimization in this study was to achieve a greater reduction in surface tension by setting the concentrations of licuri oil, glucose, ammonium nitrate, and yeast extract to optimized values. The desirability function, proposed by Derringer and Suich [36], aims to find optimal conditions for each individual variable so that the response lies within an acceptable range. In this context, each individual desirability were adjusted to achieve a global desirability index that maximized STred following Equation (7) found for the model. With a desirability index of 0.9095, the best composition for the medium was (%) licuri oil (6.2), glucose (8.45), NH_4_NO_3_ (0.62), and yeast extract (0.62), resulting in an STred of 42.782 mN·m^−1^. The behavior of the predictive values for each independent factor and desirability is represented in Figure 3.

Since the quadratic terms did not show any significant influence on the response, we could evaluate the interaction between variables through the response surface plot, providing a quicker and more direct visual analysis, as shown in Figure 4. That analysis could reveal both synergistic and antagonistic effects between the variables.

Synergistic interactions, such as the increase in response with higher levels of licuri oil and yeast extract (Figure 4C), suggest that these components may act complementarily, enhancing the desired effect. The influence of that interaction was also evidenced in the Pareto chart presented. Licuri oil likely served as an effective carbon source due to its fatty acid composition [24], which could stimulate key biosynthetic pathways involved in the production of lipid-based biosurfactants, such as glycolipids. The fatty acids in licuri oil are metabolized via β-oxidation, yielding acetyl-CoA, a critical precursor for biosurfactant synthesis [37].

On the other hand, antagonistic interactions, where the response decreases with the increase in certain pairs, indicate competition or inhibition, such as between licuri oil and glucose (Figure 4A), licuri oil and ammonium nitrate (Figure 4B), and between glucose and ammonium nitrate (Figure 4D) under certain conditions. Figure 4E,F relate the effects of interactions between glucose and ammonium nitrate with yeast extract. We can observe that in all cases, increasing the concentration of this nitrogen source to values greater than 0.3% favored the reduction in surface tension. Furthermore, the association between yeast extract and NH_4_NO_3_ at maximum concentrations achieved the highest levels of STred.

The nitrogen source also showed a significant effect on BS production by *S. marcescens*, where the combined effects of peptone (0.4%) and ammonium sulfate (0.5%) were able to reduce the surface tension to 28.4 mN·m^−1^ [34]. In the production of surfactants by *P. aeruginosa* MA01, concentrations ranged between 0.3–0.9% and 0.1–0.4% of sodium nitrate and yeast extract, respectively, for both non-optimized and optimized media, significantly increasing process yield. Yeast extract is an excellent nitrogen source for biosurfactant production due to its high bioavailability of amino acids and proteins, stimulation of metabolic pathways for secondary metabolite production, and broad compatibility with different microbial strains [38,39].

The validation of the statistical model was conducted in the subsequent stage, with the evaluation of the growth curve and determination of the optimal production time.

### 3.2. Growth Curve Analysis

Figure 5 shows the evaluation of the biomass concentration, yield, and surface tension reduction in the production of BS by *Candida mogii* during the pre-established fermentation period (216 h) in the optimized medium.

The biomass representative curve reached its maximum point (9.74 ± 0.46 g·L^−1^) within the first 96 h of fermentation, likely while there was still glucose in the medium. The low complexity of the hydrophilic source increases carbon availability, promoting its initial use by the cells as an energy source. As this source is depleted, a reduction in cell growth occurs, leading to the stationary phase, when carbon begins to be obtained from the fatty acid chains present in licuri oil. The hydrophobic nature of oils extends resistance to biodegradation due to their low solubility in water, which increases their adsorption to cell surfaces and reduces their availability for degrading microorganisms [40].

The balance between hydrophilic and hydrophobic sources in the culture medium is crucial for the quantity and characteristics of the biosurfactants (BS) produced. In this study, a 40:50 ratio was used, resulting in a maximum efficiency of 6.36 ± 0.81 g·L^−1^ at 168 h, with a productivity of 37.85 mg·L^−1^·h^−1^. Other studies have reported different optimal ratios for biosurfactant production: 10:10 (glucose/soap waste) with a rate of 49.96 mg·L^−1^·h^−1^; 100:100 (glucose/sunflower oil waste) with 216 mg·L^−1^·h^−1^; and 100:30 (glucose/rapeseed oil) with 1.64 g·L⁻^1^·h⁻^1^. These findings highlight the importance of the selection and proportion of carbon sources in the characteristics of the produced biosurfactants [41,42,43].

Although there are no records on the biosurfactant production by *C. mogii*, other species of the *Candida* genus, such as *C. bombicola* and *C. tropicalis* [12,44], have demonstrated a strong capacity to reduce surface tension. In this study, *C. mogii* achieved a reduction in surface tension from 71.042 ± 0.021 to 28.662 ± 0.563 mN·m^−1^ after 168 h in an optimized medium, highlighting its potential in biosurfactant production. Biosurfactant production began in the logarithmic phase and peaked during the decline phase, similar to what was observed in *C. catenulata* [41] and *C. tropicalis* [44], which reported a greater surface tension reduction in the stationary phase. Studies with another *C. tropicalis* strain showed higher yields already in the logarithmic phase, indicating that biosurfactant production and surface tension reduction are not necessarily linked to cellular growth [45].

### 3.3. Emulsification Index and Critical Micelle Concentration

The data on the emulsification activity of the metabolic liquid obtained from fermentation by *Candida mogii* in mineral medium supplemented with licuri oil against different hydrocarbons are presented in Table 4.

The emulsification index determines the effectiveness of the biosurfactant as a bioemulsifier. Although emulsifying and dispersing additives do not necessarily reduce the surface tension of water or hydrocarbons, they help decrease the interfacial energy between the phases [46]. To be considered a good emulsifying agent, the ability to form stable emulsions must remain above 50% for at least 24 h [47]. Among the oils tested, only canola, licuri, and motor (residual) oils showed a satisfactory emulsification activity. The values were higher than those reported by Pinto et al. (E_24_ of 59.0 ± 0.9%) [12] and Durval et al. (E_24_ of 65.8 ± 1.5%) [48] for canola oil, using *C. bombicola* and *B. cereus*, respectively. For waste engine oil, *C. mogii* showed more satisfactory results than those found by Almeida et al. using *C. tropicalis* (E_24_ close to 70%) [11].

The surface tension behavior as a function of biosurfactant concentration in the medium is shown in Figure 6. As indicated in the graph, a concentration of 0.8 g·L^−1^ was able to reduce the surface tension of the medium to the maximum value of 25.58 ± 0.31, which is the CMC for the biosurfactant produced by *C. mogii* in medium supplemented with licuri oil.

The CMC is a crucial parameter for evaluating the surface activity of a biosurfactant and its solubility in aqueous media. This parameter is influenced by the structure and composition of the biosurfactant, as well as factors such as temperature, ionic strength, and the presence of organic additives. The CMC found in this study was very close to the values achieved by *Starmerella bombicola* (0.6 g·L^−1^), *Candida utilis* (0.6 g·L^−1^), and *Yarrowia lipolytica* (1.2 g·L^−1^) reported in the literature [13,49,50].

### 3.4. Biosurfactant Characterization

The biomolecule produced by *C. mogii* exhibited an ionic charge of −98.9 mV (0.136 mS/cm, 24.9 °C), classifying it as an anionic surfactant. Similar results were reported for other *Candida* species [27,33]. A zeta potential of −98.9 mV suggested that the particles were highly stable in suspension, which is desirable in many biosurfactant applications, such as in emulsion formation or dispersion stabilization [51]. The particle size distribution by intensity showed that most particles had a size around 194.6 nm, with smaller populations of larger particles.

The isolated and purified biosurfactant was subjected to FT-IR and NMR analyses (Figure 7 and Figure 8). In the FT-IR spectrum, the band at 3402.45 cm^−1^ is associated with the hydroxyl group, indicating the presence of functional groups that may be involved in hydrogen bonding interactions. The bands at 2924.18 cm^−1^ and 2853.81 cm^−1^ correspond to the C-H stretching characteristic of alkanes, suggesting a hydrophobic structure. The presence of carbonyl groups was confirmed by the band at 1743.60 cm^−1^, which can be attributed to esters or carboxylic acids. Additionally, the band at 1620.08 cm^−1^ can be interpreted as a C=C stretching or a N-H bending, indicating the possibility of double bonds or amine groups. Finally, the band at 1408.95 cm^−1^ suggests C-H bending or C-O stretching.

The ^1^H NMR spectrum (Figure 8A) exhibited a characteristic signal at 7.28 ppm corresponding to the solvent used (deuterated chloroform). Signals between 5.20 and 5.40 ppm indicated alkene protons, suggesting unsaturation in the molecule. In the range of 4.10–5.20 ppm, signals suggested protons on carbons bonded to oxygen, as seen in esters, ethers, or vinyl protons (C=C-H). Shifts near 2.00 ppm corresponded to protons near electronegative groups, such as oxygen or nitrogen, indicating functional groups like esters or amines. Finally, the range of 0.85–1.60 ppm showed methylene and methyl protons, consistent with hydrocarbon chains.

The ^13^C NMR spectrum (Figure 8B) of the biosurfactant sample revealed several characteristic signals. A prominent peak between 77.3 and 77.5 ppm corresponded to the solvent (CDCl_3_). In the carbonyl region (170–180 ppm), signals at 172.8–173.8 ppm indicated the presence of ester or carboxylic acid groups, typical of biosurfactants derived from fatty acids. Peaks in the aromatic or alkene region (120–140 ppm), particularly between 128.5 and 129.9 ppm, suggested the presence of sp^2^ carbons, likely associated with unsaturated double bonds (C=C). The region spanning 20–60 ppm highlighted sp^3^ carbons, including methylene (-CH_2−_) groups and oxygenated carbons, such as those found in esters or glycosidic bonds (60–65 ppm). Lastly, a signal at 14.0 ppm corresponded to terminal methyl groups (-CH_3_), confirming the presence of hydrocarbon chains in the structure.

Given that the spectra showed ester groups, aliphatic chains, and oxygen, it is possible to infer that the surfactant produced by the yeast *Candida mogii* belongs to the glycolipid group, as previously reported for other species of the genus [11,52,53].

### 3.5. Phytotoxicity

The assessment of phytotoxicity in biosurfactant development is crucial to ensure environmental safety and crop health. This study confirmed that the product did not harm plants, which is essential for agricultural applications [54]. The biosurfactant toxicity was tested on lettuce seeds (*Lactuca sativa*), with the results presented in Table 5.

The results indicated that different concentrations of the biosurfactant produced by *Candida mogii* (½ CMC, 1 CMC, and 2× CMC) did not inhibit seed germination or root growth in lettuce. The observed germination indices remained above 80%, demonstrating that treatments with the biosurfactant did not negatively impact plant growth or health. Other biosurfactants also showed no negative effects on *Lactuca sativa*; for instance, mannosylerythritol lipids (MELs) at specific concentrations (158 mg·L^−1^) may even have stimulated seed germination, root growth, and development, exhibiting a biostimulant effect without phytotoxicity. Additionally, biosurfactants produced by *Bacillus subtilis* showed no toxicity towards lettuce, supporting their safe use in environmental applications [55,56].

However, while the biosurfactant was non-toxic to *Lactuca sativa*, broader environmental safety requires evaluation. Future studies should assess its effects on aquatic species, such as *Artemia salina*, and soil microbiota to ensure its suitability for diverse applications and ecological balance.

### 3.6. Antimicrobial Activity

The presence of *Staphylococcus aureus* and *Escherichia coli* in foods and environments is frequently used as an indicator of contamination and hygienic-sanitary conditions. As Gram-positive and Gram-negative bacteria, respectively, they have shown resistance to multiple antibiotics, posing an additional challenge for infection control and food safety [57,58,59]. The diameters of inhibition zones obtained from the disk diffusion test are presented in Table 6, showing a concentration-dependent inhibition zone for both strains studied.

The results presented in this study can be compared with EUCAST (The European Committee on Antimicrobial Susceptibility Testing) clinical breakpoints, which provide parameters for classifying bacterial sensitivity or resistance to conventional antimicrobials. For example, the inhibition diameters for *E. coli* at 1 CMC (12 mm) and 2 CMC (18 mm) were close to the minimum values observed for known antibiotics, such as ampicillin and cephalexin, which typically present inhibition zones between 14 and 15 mm, depending on the strain [60]. Although they did not reach the potency of broad-spectrum antibiotics, such as benzylpenicillin, which shows inhibition zones greater than 26 mm against *S. aureus*, they appear promising for specific applications, such as pathogen reduction in agro-industrial systems or contact surfaces [61].

The metabolic liquid from fermentation alone was able to inhibit microorganism growth, which is crucial since the costs associated with extraction and purification in biosurfactant production represent a disadvantage in the process [62]. Other studies have demonstrated the effectiveness of glycolipids produced by *Candida* sp. against *E. coli* and *S. aureus* [10,41,61].

Minimum Inhibitory Concentration (MIC) tests will be necessary to identify the lowest concentration of the biosurfactant capable of completely inhibiting the growth of the sample. Also, future studies may include parallel tests with standard antibiotics to validate its competitiveness and explore potential synergies between the biosurfactant and other antimicrobials.

## 4. Conclusions

The results demonstrated the potential of the yeast *Candida mogii* in the production of glycolipids, indicating promising applications in bioremediation and biotechnology. The study revealed that that strain could reduce the surface tension of water from 71.04 mN·m^−1^ to 28.66 mN·m^−1^, particularly in the configuration optimized by the quadratic model, containing (%) licuri oil (6.2), glucose (8.45), NH_4_NO_3_ (0.62), and yeast extract (0.62). The emulsification capacity of the biosurfactants produced by *C. mogii* was effective, especially concerning canola, licuri, and waste engine oils, with emulsification indices exceeding 70% after 24 h, highlighting their potential as emulsifying agents for industrial applications. The developed polynomial model showed a good fit to the experimental data, with R^2^ and adjusted R^2^ values of 0.9451 and 0.8812, respectively, ensuring the model’s reliability in predicting and optimizing biosurfactant production parameters. The biosurfactant exhibited no toxicity towards *Lactuca sativa* seeds and was identified as a promising antimicrobial agent against *S. aureus* and *E. coli*.

Further research is essential to confirm the bioproduct’s safety for a broader range of organisms, strengthen the evidence of its antimicrobial efficacy through direct comparisons with standard antibiotics, and investigate its antibiofilm potential and underlying mechanisms of action. Future studies should also focus on scaling up production processes, testing the biosurfactant in contaminated environments, and conducting field trials to validate its effectiveness in real-world applications, ensuring its feasibility for industrial and environmental use.

## Figures and Tables

**Figure 1 foods-13-04029-f001:**
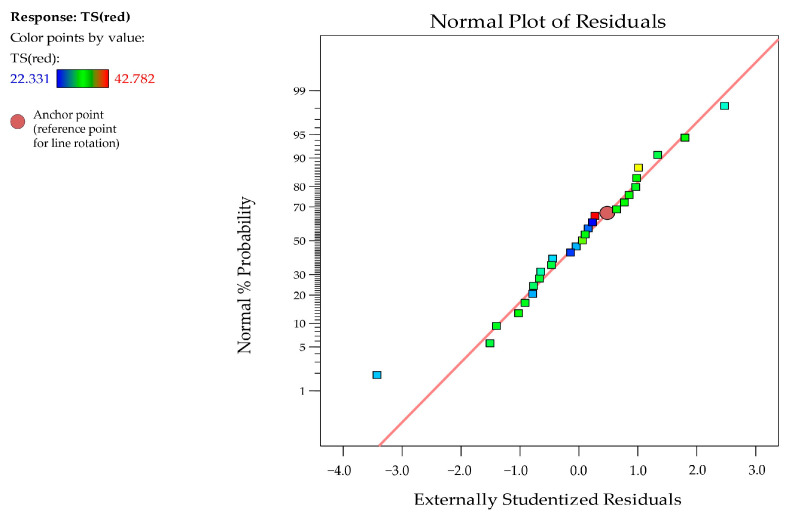
Normal plot of residuals indicating the normal distribution of the model applied to optimize the reduction in surface tension by *C. mogii*.

**Figure 2 foods-13-04029-f002:**
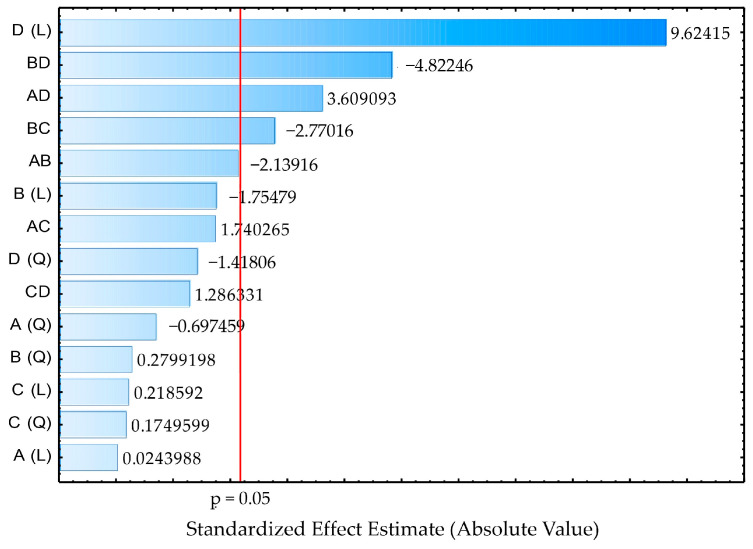
Pareto chart for STred according to the statistical analysis of the CCD carried out to evaluate the effect of independent variables’ concentration in the culture medium for the biosurfactant production.

**Figure 3 foods-13-04029-f003:**
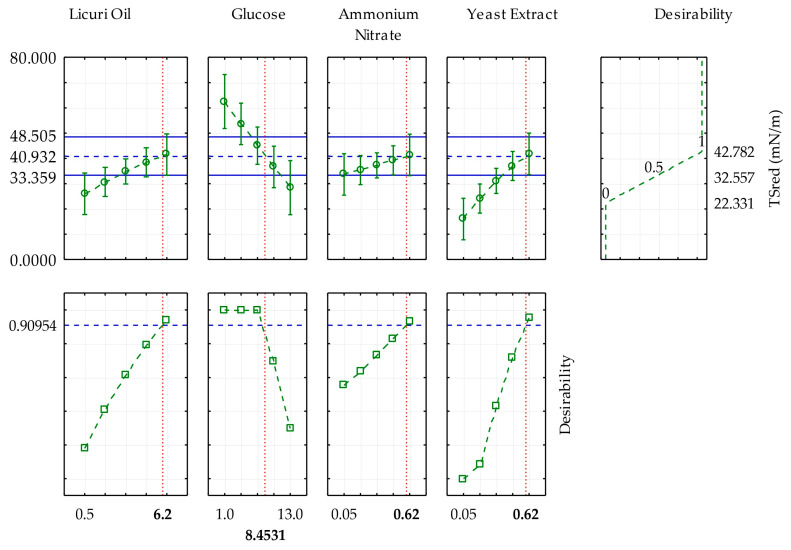
Profiles for predicted values and desirability for ST_red_ based on the statistical analysis of the CCD adopted in this study.

**Figure 4 foods-13-04029-f004:**
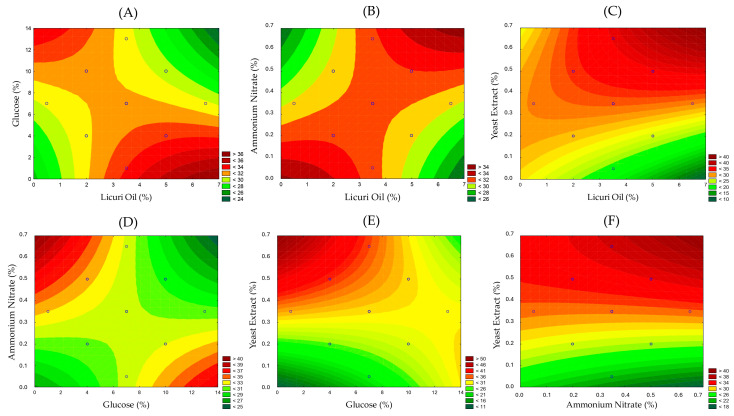
Response profiles for the interactions between variables: (**A**) licuri oil and glucose, (**B**) licuri oil and ammonium nitrate, (**C**) licuri oil and yeast extract, (**D**) glucose and ammonium nitrate, (**E**) glucose and yeast extract, and (**F**) ammonium nitrate and yeast extract.

**Figure 5 foods-13-04029-f005:**
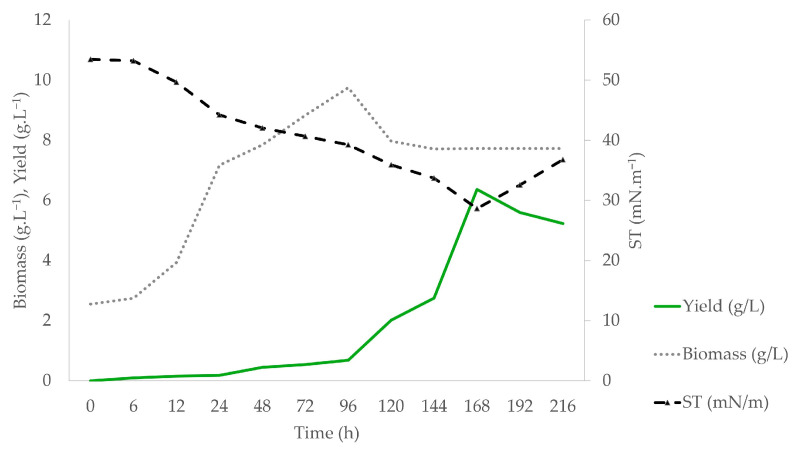
Temporal changes in biomass, yield and surface tension during cultivation of *C. mogii* in mineral medium.

**Figure 6 foods-13-04029-f006:**
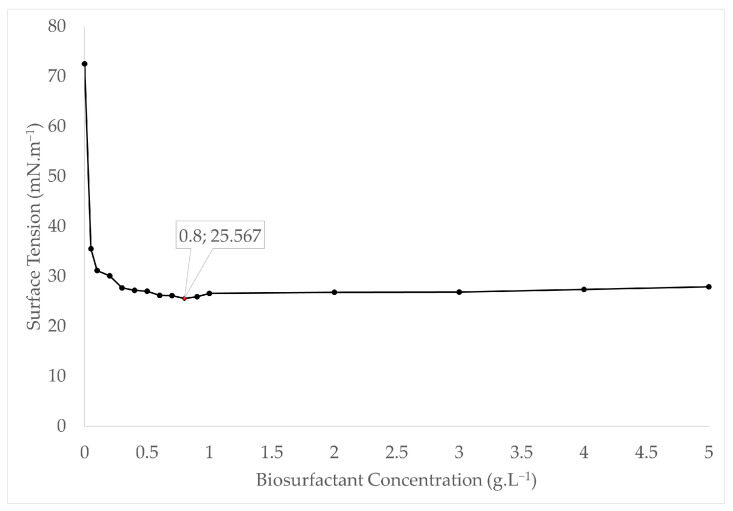
Critical micelle concentration of the biosurfactant from *C. mogii*.

**Figure 7 foods-13-04029-f007:**
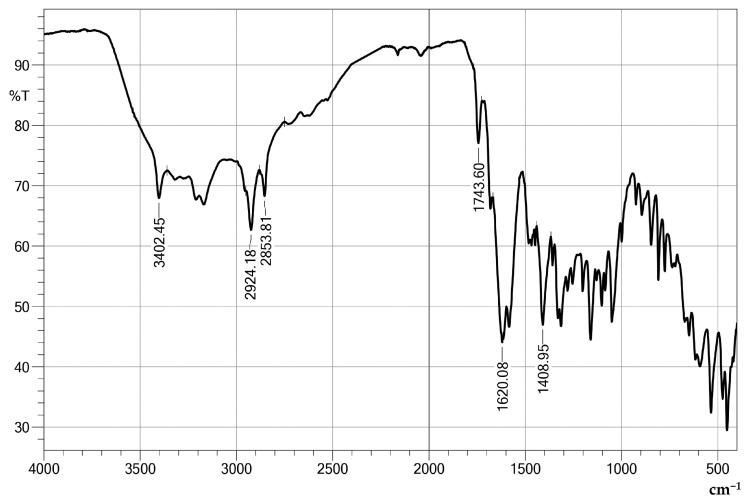
FT-IR spectrum of biosurfactant produced by *C. mogii* grown in an optimized medium containing licuri oil.

**Figure 8 foods-13-04029-f008:**
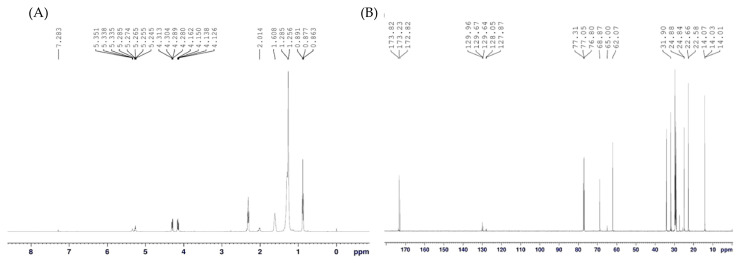
NMR spectrum of biosurfactant produced by *C. mogii* grown in an optimized medium containing licuri oil: (**A**) ^1^H and (**B**) ^13^C.

**Table 1 foods-13-04029-t001:** Values of the input variables used in the Central Composite Rotational Design (CCRD).

Variables	Levels				
	−1.41	−1	0	+1	+1.41
A (%)	0.50	2.00	3.50	5.00	6.50
B (%)	1.00	4.00	7.00	10.0	13.0
C (%)	0.05	0.20	0.35	0.50	0.65
D (%)	0.05	0.20	0.35	0.50	0.65

**Table 2 foods-13-04029-t002:** Central Composite Rotational Design (CCRD) with observed and predicted STred.

Run	A (%)	B (%)	C (%)	D (%)	STexp	STred (Observed)	STred (Predicted)
1	5.00	4.00	0.50	0.20	45.244	25.783	26.564
2	2.00	10.0	0.20	0.20	35.603	34.921	34.065
3	2.00	4.00	0.50	0.20	42.707	28.472	26.428
4	2.00	4.00	0.20	0.50	37.063	32.576	31.603
5	3.50	7.00	0.35	0.35	36.258	30.321	31.291
6	3.50	13.0	0.35	0.35	40.081	32.145	30.512
7	3.50	7.00	0.35	0.35	44.847	31.438	31.291
8	2.00	10.0	0.20	0.50	40.347	31.098	31.768
9	5.00	10.0	0.20	0.50	47.587	31.265	32.167
10	5.00	4.00	0.20	0.20	33.577	23.592	23.732
11	6.50	7.00	0.35	0.35	45.396	31.254	30.296
12	5.00	4.00	0.50	0.50	28.397	42.782	42.496
13	2.00	4.00	0.50	0.50	44.044	34.116	35.119
14	5.00	4.00	0.20	0.50	39.914	37.602	36.601
15	3.50	7.00	0.05	0.35	46.688	30.924	31.393
16	2.00	10.0	0.50	0.20	40.258	26.332	28.784
17	0.50	7.00	0.35	0.35	42.068	29.111	29.760
18	3.50	1.00	0.35	0.35	39.925	31.265	32.589
19	3.50	7.00	0.35	0.35	39.914	32.115	31.291
20	5.00	10.0	0.50	0.50	39.034	30.921	32.33
21	5.00	10.0	0.50	0.20	40.885	24.491	24.323
22	5.00	10.0	0.20	0.20	38.39	26.775	27.223
23	3.50	7.00	0.35	0.05	48.848	22.331	22.088
24	2.00	4.00	0.20	0.20	35.39	25.935	25.976
25	3.50	7.00	0.35	0.65	39.064	35.789	35.722
26	3.50	7.00	0.65	0.35	40.585	32.786	32.008
27	2.00	10.0	0.50	0.50	39.741	30.832	29.550

The coded variables represent the concentrations of licuri oil (A), glucose (B), ammonium nitrate (C), and yeast extract (D). STexp denotes the experimental value obtained during the measurement of surface tension using the tensiometer. STred denotes the surface tension reduction, calculated as the difference between the surface tension of water (71.02 mN·m^−1^) and STexp in each configuration.

**Table 3 foods-13-04029-t003:** ANOVA analysis of the quadratic model for surface tension reduction.

Source	Sum of Squares	df	Mean Square	F-Value	*p*-Value
Model	482.70	14	34.478	14.775	<0.0001 ^s^
A—Licuri oil	0.43068	1	0.43068	0.18456	0.6751 ^ns^
B—Glucose	6.4719	1	6.4719	2.7734	0.1217 ^ns^
C—Ammonium nitrate	0.56703	1	0.56703	0.24299	0.6309 ^ns^
D—Yeast extract	278.85	1	278.85	119.50	<0.0001 ^s^
AB	21.139	1	21.139	9.0588	0.0108 ^s^
AC	5.6656	1	5.6656	2.4279	0.1451 ^ns^
AD	52.443	1	52.443	22.473	0.0004 ^s^
BC	32.864	1	32.864	14.083	0.0027 ^s^
BD	62.794	1	62.794	26.909	0.0002 ^s^
CD	9.3866	1	9.3866	4.0224	0.0679 ^ns^
A^2^	2.1293	1	2.1293	0.91246	0.3583 ^ns^
B^2^	0.089298	1	0.089298	0.038267	0.8481 ^ns^
C^2^	0.22281	1	0.22281	0.095483	0.7626 ^ns^
D^2^	7.5920	1	7.5920	3.2534	0.0964 ^ns^
Lack of fit	26.361	10	2.6361	3.2119	0.26069 ^ns^
Pure error	1.6415	2	0.82074		
R^2^ = 0.9451; R^2^_adj_ = 0.8812; adeq. precision = 17.924					

^s^ indicates significant difference at *p* < 0.05. ^ns^ indicates no significant difference at *p* > 0.05.

**Table 4 foods-13-04029-t004:** Emulsification index (E_24_) for BS produced by *C. mogii* against various hydrocarbons.

Hydrocarbon	E_24_ (%)	Hydrocarbon	E_24_ (%)
Canola oil	75.43 ± 0.61	Waste engine oil	80.48 ± 7.74
Licuri oil	82.38 ± 15.49	Diesel	8.81 ± 0.74
Soybean oil	5.79 ± 0.13	Kerosene	3.28 ± 0.05

Experiments were performed in triplicate, and the results represent the mean ± standard deviation of two independent experiments.

**Table 5 foods-13-04029-t005:** Biosurfactant toxicity test on *Lactuca sativa* seeds.

Concentration	Germination	Root Growth	GI
1/2 CMC	89.29% ^a^	99.73% ^a^	89.05% ^a^
CMC	92.86% ^a^	95.88% ^a^	89.03% ^a^
2× CMC	92.86% ^a^	89.01% ^b^	82.65% ^b^

GI: germination index. Data with different indices (^a^ and ^b^) indicate a statistically significant difference within the same column.

**Table 6 foods-13-04029-t006:** The average diameter of inhibition zone against *S. aureus* and *E. coli* in response to the biosurfactant produced by *C. mogii* in medium supplemented with licuri oil.

Sample	Inhibition Zone of Growth (mm)	
	*S. aureus*	*E. coli*
Metabolic liquid	13	8
^1^/_2_ CMC	6	6
1 CMC	15	12
2 CMC	16	18
Negative control	-	-

## Data Availability

The original contributions presented in the study are included in the article. Further inquiries can be directed to the corresponding author.

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
