# Peer review of "Production and Optimization of Biosurfactant Properties Using Candida mogii and Licuri Oil (Syagrus coronata)"

_foods, 2024, doi:10.3390/foods13244029_

Round 1

Reviewer 1 Report

Comments and Suggestions for Authors

Some of my comments that needs to be adressed for improvement of the manuscript:

1. The introduction needs to more clearly connect the unique benefits of Candida mogii with prior studies on similar yeast strains. For example, how does its performance compare with other Candida species, such as C. bombicola or C. tropicalis?

2. While the manuscript identifies the optimal substrate composition, it does not sufficiently explore the biological or chemical mechanisms underlying these optimizations. Why do specific concentrations of licuri oil and yeast extract result in the greatest reduction in surface tension?

3. Although the manuscript highlights the biosurfactant's non-toxicity to Lactuca sativa, its potential toxicity to other organisms, such as aquatic species or soil microbes, should be evaluated to support broader environmental applications. A discussion on this would be valuable.

4. Replace "CFU.ml-1" with "CFU/mL."

5. Ensure microbial names are italicized. For instance, review Line 92 for this correction.

6. The manuscript reports inhibition zones for Staphylococcus aureus and Escherichia coli in response to the biosurfactant but does not sufficiently interpret these findings:

a. The reported inhibition zones (e.g., 15 mm for S. aureus and 12 mm for E. coli at 1 CMC concentration) are presented without context. How do these values compare to standard antibiotics or other antimicrobial agents?

b. It is unclear whether the inhibition zones meet thresholds for industrial relevance.

c. The relationship between the biosurfactant's properties (e.g., critical micelle concentration, ionic charge) and its antimicrobial activity is not addressed, leaving the findings disconnected from the broader discussion.

d. The antimicrobial activity against Staphylococcus aureus and Escherichia coli is demonstrated, but the manuscript lacks insight into the mechanisms behind this effect. Discussing potential modes of action, such as membrane disruption or enzymatic inhibition, would enhance the antimicrobial claims.

I recommend expanding the discussion to include a more thorough interpretation of the antimicrobial activity of the biosurfactant.

7. While the manuscript concludes with promising applications, it does not outline specific future experiments, such as scaling up production, testing in contaminated environments, or conducting field trials. Including these aspects in the conclusion would provide a clearer direction for future research.

Author Response

Comments 1: “The introduction needs to more clearly connect the unique benefits of Candida mogii with prior studies on similar yeast strains. For example, how does its performance compare with other Candida species, such as C. bombicola or C. tropicalis?”

Response 1: Thank you very much for pointing this out. Indeed, the comparison was not conducted because there are no reports in the literature indicating the use of C. mogii for this purpose. However, the aim of this study is precisely to introduce, within the Candida genus, a new strain with potential for surfactant production. Modifications have been made to the introduction to highlight (lines 52–55) the yeast’s applications for other purposes and emphasize its potential as a candidate for biosurfactant production.

Comments 2: “While the manuscript identifies the optimal substrate composition, it does not sufficiently explore the biological or chemical mechanisms underlying these optimizations. Why do specific concentrations of licuri oil and yeast extract result in the greatest reduction in surface tension?”

Response 2: Agreed. Paragraphs have been added/modified (lines 298–305 and 320–323) in the discussion to explore how the increase in these variables can contribute to enhanced biosurfactant production, which, in turn, would result in greater surface tension reduction.

Comments 3: “ Although the manuscript highlights the biosurfactant's non-toxicity to Lactuca sativa, its potential toxicity to other organisms, such as aquatic species or soil microbes, should be evaluated to support broader environmental applications. A discussion on this would be valuable.”

Response 3: Thank you very much for this consideration. Indeed, the absence of phytotoxicity does not imply non-toxicity to other organisms. Further assays with other bioindicators (marine and terrestrial) are necessary to confirm this safety. The text has been revised to include suggestions for future analyses (lines 456–459).

Comments 4: “Replace "CFU.ml-1" with "CFU/mL."”

Response 4: Thank you for the suggestion, but the units were presented in accordance with the International System, which states that both formats (with a slash or exponent) are acceptable.

Comments 5: ‘Ensure microbial names are italicized. For instance, review Line 92 for this correction.”

Response 5: Thank you for pointing that out. The error has been corrected.

Comments 6: “The manuscript reports inhibition zones for Staphylococcus aureus and Escherichia coli in response to the biosurfactant but does not sufficiently interpret these findings:

  1. The reported inhibition zones (e.g., 15 mm for S. aureus and 12 mm for E. coliat 1 CMC concentration) are presented without context. How do these values compare to standard antibiotics or other antimicrobial agents?
  2. It is unclear whether the inhibition zones meet thresholds for industrial relevance.
  3. The relationship between the biosurfactant's properties (e.g., critical micelle concentration, ionic charge) and its antimicrobial activity is not addressed, leaving the findings disconnected from the broader discussion.
  4. The antimicrobial activity against Staphylococcus aureusand Escherichia coli is demonstrated, but the manuscript lacks insight into the mechanisms behind this effect. Discussing potential modes of action, such as membrane disruption or enzymatic inhibition, would enhance the antimicrobial claims.

I recommend expanding the discussion to include a more thorough interpretation of the antimicrobial activity of the biosurfactant.”

Response 6: Agreed. The section has been improved (lines 469–483 and 487–489) to compare the inhibition diameters found with the Clinical Breakpoint Table of the European Committee on Antimicrobial Susceptibility Testing. This allows for contextualizing the biosurfactant's effectiveness in relation to conventional antimicrobials, providing a more robust view of its practical relevance. Additionally, it suggests future studies that include parallel tests with standard antibiotics to validate its competitiveness and explore potential synergies between the biosurfactant and other antimicrobials.

Furthermore, there is no direct relationship between the biosurfactant’s properties (such as CMC or ionic charge) and its antimicrobial potential. The determinations were made to characterize the biomolecule (ionic charge) and display its working concentration (CMC).

Comments 7:  While the manuscript concludes with promising applications, it does not outline specific future experiments, such as scaling up production, testing in contaminated environments, or conducting field trials. Including these aspects in the conclusion would provide a clearer direction for future research.

Response 7: Agreed. Suggestions for future work have been added (lines 504–510), aiming to continue the research on biosurfactants produced by Candida mogii, proposing studies that enhance both characterization and potential applications.

Reviewer 2 Report

Comments and Suggestions for Authors

The manuscript entitled “Biosurfactant with antimicrobial potential produced by Candida mogii on an optimized substrate supplemented with licuri oil (Syagrus coronata), describes the optimization of the biosurfactant production medium for Candida mogii using licuri oil as an alternative carbon source and to evaluate its potential as an antimicrobial agent. ​

Although the manuscript interesting and mostly well organized, and the research fine, there are some points that need to be underlined.

-          The overall manuscript needs to be checked in terms of grammar and consistency before submission.

-          (line 57, please add the abbreviation CCD).

-          The scope is almost clear

-          The figures are fine, and the tables can be improved.

Title

-          It can be improved. I would suggest following title “Production and Optimization of Biosurfactant Properties Using Candida mogii and Licuri Oil (Syagrus coronata)

Introduction

-          It is too short. More information covering all aspects is needed.

-          It can be improved

M&M

-          correct the spelling of Materiais“

-          2.3. it better to replace it to “Yeast inoculation”

-          Table 1 should be moved to result section

-          2.5. the sub-heading should be “Surface tension measurement”.

-          2.6. where does this method comes from? Please add the reference/s in this section

-           2.7. use abbreviations (EI) after sub-heading.

-          Line (143-145): It is recommended to clarify that the higher EI percentage means more stable emulsion and vice versa.

-          Line (192): Use italic font for microorganisms

-          Line (194): It is better to express the concentration as g.L⁻¹

-          Line (195, 197): will be ?? use past tense.

-          Line (196) Muller-Hinton: provide the name and place of manufacture.

Results

-          Table:2.  Use Symbols in the table and explain them in table footnote.

-          Line (213-214): please revise this sentence to be clearer. I would suggest using the following sentence "The effectiveness of a biosurfactant is determined by its ability to reduce surface tension; a greater reduction indicates higher effectiveness.

-          Line (250) Figure 2. Fix it or remove it.

-          Line (313). No need for repeating “UFPEDA 3968”

Conclusions

-          Can be improved 

Author Response

The manuscript entitled “Biosurfactant with antimicrobial potential produced by Candida mogii on an optimized substrate supplemented with licuri oil (Syagrus coronata), describes the optimization of the biosurfactant production medium for Candida mogii using licuri oil as an alternative carbon source and to evaluate its potential as an antimicrobial agent. ​

Although the manuscript interesting and mostly well organized, and the research fine, there are some points that need to be underlined.

Comments 1: “The overall manuscript needs to be checked in terms of grammar and consistency before submission.”

Response 1: Thank you for mentioning this. The entire text has been re-evaluated for cohesion, coherence, and grammar. The changes can be found in red throughout the manuscript.

Comments 2: “(line 57, please add the abbreviation CCD).”

Response 2: Thank you for pointing that out. The change has been made (line 60).

Comments 3: “The scope is almost clear”

Response 3: Agreed. Changes have been made in the introduction (lines 66–70 and 73–77) to emphasize the use of Licuri as a substrate and the objective of the work.

Comments 4: “The figures are fine, and the tables can be improved.”

Response 4: Agreed. All tables have been revisited to improve clarity.

Comments 5: “Title

-          It can be improved. I would suggest following title “Production and Optimization of Biosurfactant Properties Using Candida mogii and Licuri Oil (Syagrus coronata)”

Response 5: Thank you for pointing that out. We believe the revision highlights the objective of the study, which is to optimize the surface tension reduction property of the biosurfactant produced by Candida mogii in a medium supplemented with licuri oil (Syagrus coronata).

Comments 6: “Introduction

-          It is too short. More information covering all aspects is needed.

-          It can be improved”

Response 6: Agreed. Information about the yeast Candida mogii and the lack of its applications in biosurfactant production has been added (lines 52–55). The text has also been supplemented with relevant details on Licuri oil (lines 66–71) and the objective of the work (lines 73–77).

Comments 7: “M&M

-          correct the spelling of Materiais“

-          2.3. it better to replace it to “Yeast inoculation”

Response 7:Agreed. Changes have been made (lines 79 and 90).

Comments 8: “Table 1 should be moved to result section”

Response 8: Thank you for pointing that out. Table 1 is located in the Materials and Methods section as it presents the compositions used in the development of the experimental design. Since it does not necessarily present a result, we believe it is appropriately placed in this section.

Comments 9:-          2.5. the sub-heading should be “Surface tension measurement”.

Response 9: Agreed. The correction has been made (line 129).

Comments 10 “-          2.6. where does this method comes from? Please add the reference/s in this section”

Response 10: Thank you for pointing that out. The reference has been added to the section (line 142).

Comments 11 - ”          2.7. use abbreviations (EI) after sub-heading.”

Response 11: Thank you, I understand your point, but the abbreviation E24 is widely used to refer to the emulsification index, including in the reference cited (Cooper and Goldberg, 1987).

Comments 12: “-          Line (143-145): It is recommended to clarify that the higher EI percentage means more stable emulsion and vice versa.”

Response 12: Agreed. The explanation has been added in lines 152–153.

Comments 13: “-          Line (192): Use italic font for microorganisms”

Response 13: Thank you for pointing that out. The item has been corrected (line 200).

Comments 13: “-          Line (194): It is better to express the concentration as g.L⁻¹”

Response 13: Thank you for pointing that out. The correction is in line 202.

Comments 14: “-          Line (195, 197): will be ?? use past tense.

-          Line (196) Muller-Hinton: provide the name and place of manufacture.”

Response 14: Agreed. Modifications have been made in the sections and are highlighted in red in the text.

Comments 15: “Results

-          Table:2.  Use Symbols in the table and explain them in table footnote.

-          Line (213-214): please revise this sentence to be clearer. I would suggest using the following sentence "The effectiveness of a biosurfactant is determined by its ability to reduce surface tension; a greater reduction indicates higher effectiveness.

-          Line (250) Figure 2. Fix it or remove it.

-          Line (313). No need for repeating “UFPEDA 3968””

Response 15: Agreed. Footnotes have been added to Table 2 (lines 223–228) to explain the variables present in the table. The sentence "The effectiveness of a biosurfactant is determined by its ability to reduce surface tension; a greater reduction indicates higher effectiveness." was accepted as a suggestion to replace the previous one (lines 229–230). The Pareto chart (Figure 2) has been adjusted to make the information clearer. The term UFPEDA 3968 has been removed, as suggested.

 Comments 16: “Conclusions

 -          Can be improved”

Response 16: Agreed. The conclusion has been improved with the addition of suggestions for future work (lines 504–510) aimed at continuing research on biosurfactants produced by Candida mogii, proposing studies that enhance both characterization and potential applications.

Reviewer 3 Report

Comments and Suggestions for Authors

Correct the language of figure 5 - Time instead Tempo

Figure 8 needs peak designations.

Correct the references according to the journal requirements.

Author Response

Comments 1: “Correct the language of figure 5 - Time instead Tempo”

Response 1: Thank you for pointing that out. The word has been changed in Figure 5.

Comments 2: “Figure 8 needs peak designations.”

Response 2: Agreed. As requested, the figure has been modified. Additionally, the text preceding it has been improved to enhance clarity (lines 414–431).

Comments 3: “Correct the references according to the journal requirements.”

Response 3: You're very welcome. All references have been reviewed and are now presented according to the format outlined by the journal in the template.